# Deciphering minimal antigenic epitopes associated with *Burkholderia pseudomallei* and *Burkholderia mallei* lipopolysaccharide O-antigens

Marielle Tamigney Kenfack[1], Marcelina Mazur[1,2], Teerapat Nualnoi[3,4], Teresa L. Shaffer[5], Abba Ngassimou[1], Yves Blériot[1], Jérôme Marrot[6], Roberta Marchetti[7], Kitisak Sintiprungrat[8], Narisara Chantratita[8,9], Alba Silipo[7], Antonio Molinaro[7], David P. AuCoin[3], Mary N. Burtnick[5], Paul J. Brett[5] & Charles Gauthier[1,10]

*Burkholderia pseudomallei* (*Bp*) and *Burkholderia mallei* (*Bm*), the etiologic agents of melioidosis and glanders, respectively, cause severe disease in both humans and animals. Studies have highlighted the importance of *Bp* and *Bm* lipopolysaccharides (LPS) as vaccine candidates. Here we describe the synthesis of seven oligosaccharides as the minimal structures featuring all of the reported acetylation/methylation patterns associated with *Bp* and *Bm* LPS O-antigens (OAgs). Our approach is based on the conversion of an L-rhamnose into a 6-deoxy-L-talose residue at a late stage of the synthetic sequence. Using biochemical and biophysical methods, we demonstrate the binding of several *Bp* and *Bm* LPS-specific monoclonal antibodies with terminal OAg residues. Mice immunized with terminal disaccharide–CRM197 constructs produced high-titer antibody responses that crossreacted with *Bm*-like OAgs. Collectively, these studies serve as foundation for the development of novel therapeutics, diagnostics, and vaccine candidates to combat diseases caused by *Bp* and *Bm*.

[1] Institut de Chimie IC2MP, CNRS-UMR 7285, Équipe Synthèse Organique, Groupe Glycochimie, Université de Poitiers, 4, rue Michel Brunet, Poitiers 86073, France. [2] Department of Chemistry, Wroclaw University of Environmental and Life Sciences, C. K. Norwida 25, Wroclaw 50-375, Poland. [3] Department of Microbiology and Immunology, University of Nevada School of Medicine, 1664, N. Virginia Street, Reno, Nevada 89557, USA. [4] Department of Pharmaceutical Technology, Faculty of Pharmaceutical Sciences, Prince of Songkla University, 15, Kanjanavanit Road, 90112 Songkhla, Thailand. [5] Department of Microbiology and Immunology, University of South Alabama, 610, Clinic Drive, Mobile, Alabama 36688, USA. [6] Institut Lavoisier de Versailles, CNRS-UMR 8180, Université de Versailles Saint-Quentin-en-Yvelines, Université Paris-Saclay, 45, Avenue des États-Unis, Versailles 78035, France. [7] Department of Chemical Sciences, Università di Napoli Federico II, Complesso Universitario Monte S. Angelo, Via Cintia 4, Naples I-80126, Italy. [8] Department of Microbiology and Immunology, Faculty of Tropical Medicine, Mahidol University, 420/6 Rajvithi Road, Bangkok 10400, Thailand. [9] Mahidol-Oxford Tropical Medicine Research Unit, Faculty of Tropical Medicine, Mahidol University, 420/6 Rajvithi Road, Bangkok 10400, Thailand. [10] INRS-Institut Armand-Frappier, Université du Québec, 531, Boulevard des Prairies, Laval (Québec), Canada H7V 1B7. Correspondence and requests for materials should be addressed to P.J.B. (email: pbrett@southalabama.edu) or to C.G. (email: charles.gauthier@iaf.inrs.ca)

**B**urkholderia pseudomallei (Bp) is the causative agent of melioidosis, a multifaceted tropical disease leading to death in up to 50% of infected patients[1–3]. The genetically related Burkholderia mallei (Bm), the causative agent of glanders, primarily infects solipeds but can also lead to fatal infections in humans if left untreated[4]. These facultative intracellular, Gram-negative bacteria are both CDC Tier 1 select agents because of their high infectivity via inhalation, low infectious doses, and potential for misuse as biothreat agents, especially in the aerosolized form[5]. There are no clinically approved prophylactic vaccines currently available for either of these infections, thus the development of effective countermeasures is of outmost importance to combat disease caused by these bacterial pathogens[6–14].

Bp and Bm produce structurally similar lipopolysaccharides (LPS) anchored in their outer membranes. Bp and Bm LPS are potent activators of human Toll-like receptor 4[15, 16], stimulate human macrophage-like cells[15], are important virulence factors[17–19], and play a central role in host–pathogen interactions[20, 21]. Importantly, levels of anti-LPS antibodies are significantly higher in melioidosis patients who survive in comparison to those who succumb to disease[22]. Additionally, LPS-specific monoclonal antibodies (mAbs) have been shown to be passively protective in animal models of infection[23–28]. Several studies have highlighted the potential of Bp and Bm LPS as subunit vaccine candidates for melioidosis and glanders. Mice immunized with LPS from Bp, and from the non-pathogenic Burkholderia thailandensis (Bt), developed high-titer immunoglobulin G (IgG) responses and were partially protected against lethal challenges of Bp[29, 30]. In recent years, glycoconjugate vaccines composed of LPS (or detoxified LPS) covalently linked to carrier proteins and/or gold nanoparticles have been evaluated in mice and non-human primates with promising results according to their immunogenicity and protective efficacy[31–38].

Structurally, Bp and Bm LPS antigens comprise three distinct domains (e.g., lipid A[39], inner and outer core, and the O-antigen (OAg) repeat) (Fig. 1). The OAg structure consists of a linear heteropolymer featuring a disaccharide as the repeating unit in an equimolar ratio of (1→3)-linked 6-deoxy-α-L-talopyranose and β-D-glucopyranose[40–42]. Interspecies variations within the OAg lie in the different substitutions of the 6-deoxytalose residues, e.g., O-acetylation at both C4 and C2 and O-methylation at C2[43]. We have recently revisited the acetylation and methylation patterns of Bp, Bm, and Bt OAg and found that five intrachain (internal, **A–E**) as well as two terminal (non-reducing, **F** and **G**) disaccharides occur in variable proportions within the OAg (Fig. 1)[44, 45]. Although O-acetylation at the C4 position has been detected in significant amounts in Bp, Bm strains do not incorporate this modification. Moreover, as another atypical characteristic of these OAgs, the terminal residues at the non-reducing end are decorated with a methyl group at the C3 position. It has been shown that differences in colony morphology (mucoid vs non-mucoid strains of Bp) are associated with OAg substitution patterns, which influence interactions with LPS-specific mAbs[46]. We have hypothesized that these different OAg modifications could have profound impact for antibody recognition and immune responses[47], and therefore are crucial structural parameters to take into consideration for the development of LPS-based vaccines against Bp and Bm.

For the first time, we describe an efficient synthetic approach allowing access to seven oligosaccharides (**1–7**) featuring all of the reported intrachain (trisaccharides **1–5**) and terminal (disaccharides **6** and **7**) epitopes of Bp and Bm OAg. The synthetic routes and target compounds were devised in order to avoid potential acetyl migration on the all cis-triol 6-deoxytalose residue. Molecular interactions of the synthetic oligosaccharides with Bp and Bm LPS-specific mAbs were probed using enzyme-linked immunosorbent assay (ELISA) glycan arrays, surface plasmon resonance (SPR), and saturation transfer

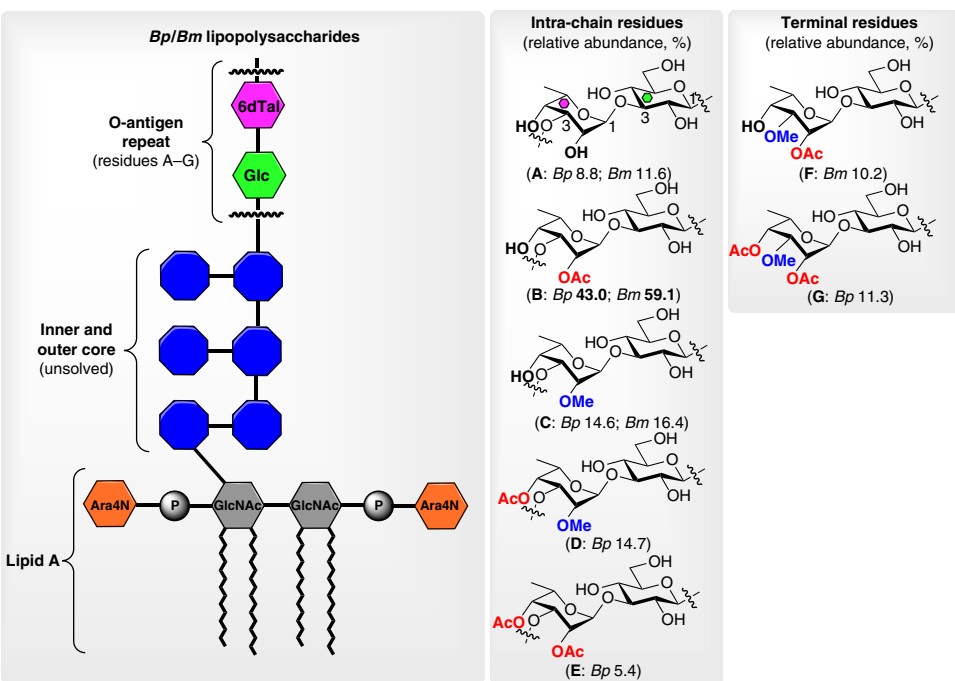

**Fig. 1** Chemical structure of B. pseudomallei and B. mallei LPS antigens. Smooth LPS species consist of three major domains: the lipid A, the core, and the OAg repeat. The OAg is a linear heteropolymer featuring a disaccharide unit in an equimolar ratio of (1→3)-linked 6-deoxy-α-L-talopyranose and β-D-glucopyranose. Five internal (intrachain) and two terminal (non-reducing) disaccharide residues are present within the OAg. According to the species, they show different methylation and acetylation substitution patterns at the C2, C3, and C4 positions of the 6-deoxy-L-talose residue[45]

**Fig. 2** Planned retrosynthetic analysis of the target oligosaccharides **1–7**. *Ac* acetyl, *Bn* benzyl, *Lev* levulinoyl, *Ph* phenyl, *PMB para*-methoxybenzyl, *SEt* thioethyl, *TBS tert*-butyldimethylsilyl

difference (STD)-nuclear magnetic resonance (NMR). We show that the mAbs strongly interact with the 6-deoxytalose residue of the 3-O-methylated terminal disaccharides. Based on these results, the two terminal disaccharides **6** and **7** were covalently linked to CRM197 carrier protein and evaluated in mice for their immunogenicity. High-titer antibody responses were raised against disaccharide **6** of the constructs, and these responses were crossreactive with *Bm*-like LPS. Collectively, these studies represent a novel platform for the development of glycoconjugate vaccines and diagnostics to combat melioidosis and glanders.

## Results

**Synthetic approach**. The target compounds **1–7** were conceived as the shortest possible oligosaccharides mimicking the substitution epitopes of the 6-deoxytalose residue, without anticipated acetyl migration. We first planned to introduce the *O*-acetyl and *O*-methyl groups on the talose unit prior to its incorporation into the oligosaccharides. Thus, according to the retrosynthetic analysis depicted in Fig. 2, the target oligo-saccharides would come from five suitably functionalized talose donors (**8–12**), which are activated at their anomeric position with a trichloroacetimidate (TCA) group[48]. The choice of the TCA group was motivated by the high-yielding coupling reported for structurally similar L-rhamnose donors in the context of the synthesis of bacterial glycans[49]. All of these donors (**8–12**) were synthesized using a C4 oxidation/reduction sequence from a common allylated rhamnose precursor followed by subsequent regioselective 3-O-methylation or 3-O-*para*-methoxybenzylation via optimization of the stannylene acetal chemistry[50] (Supplementary Figs. 1 and 2 and Supplementary Table 1). The *para*-methoxybenzyl (PMB) group would allow, once deprotected, the introduction of the terminal glucose moiety at the C3 position while the benzyl (Bn) group would act as a permanent blocker of the C4 position for donors **8** and **11**. The glucose residue at the reducing end, i.e., acceptor **13** (Supplementary Fig. 3), is functionalized with an aliphatic azidolinker chain, which would allow its transformation into a

primary amine upon hydrogenolysis. This amine would serve as an anchor for subsequent biotinylation and covalent coupling with a carrier protein. Thioglycoside donor **14** (Supplementary Fig. 4) was conceived for the introduction of the terminal glucose unit. It bears a levulinoyl (Lev) group at C2, which would act as a neighboring participating group for the formation of the 1,2-*trans*-linkage in addition to being orthogonal to acetyl groups. The presence of a *tert*-butyldimethylsilyl (TBS) group at C4 would allow the synthesis of longer oligosaccharide chains upon deprotection. Furthermore, if the coupling proves unsuccessful with thioglycoside **14**, the latter would be readily convertible into other donors, such as anomeric fluorides and imidates.

**Synthesis of protected disaccharides**. Disaccharides **15–19** were prepared from TCA talose donors **8–12** and acceptor **13** under the catalytic promotion of trimethylsilyl tri-fluoromethanesulfonate (TMSOTf) at –10 °C. Optimization of the glycosylation reactions was first performed with donor **8** (entries 1–4, Table 1) by varying the solvent, reaction time, equivalents of TMSOTf, and the presence or absence of water-scavenging 4 Å molecular sieves (MS). When conducting the glycosylation in 1,2-dichloroethane (DCE) in the presence of MS (entry 1), desired disaccharide **15** was obtained in poor yield (30%) along with disaccharide **20** as the major compound resulting from the cleavage of the PMB under catalytic acid conditions, which was somewhat unexpected for this protecting group. Interestingly, reacting disaccharide **15** under TMSOTf-catalyzed conditions led to a complex mixture of degradation products while no disaccharide **20** was observed. Loss of the PMB during the course of the glycosylation reaction could thus be rationalized by the steric effect of the Bn group at C4 combined with the electron-donating properties of the PMB group at C3 (Supplementary Fig. 5). Indeed, the dioxalenium ion could be attacked by the C3 oxygen atom leading to PMB cleavage together with the formation of a 1,2,3-*O*-orthoacetyl species. Once activated by TMSOTf, this tricyclic orthoester could be converted into the thermo-dynamically favored alcohol **20** upon attack of acceptor **13**.

**Table 1 Synthesis of protected disaccharides**

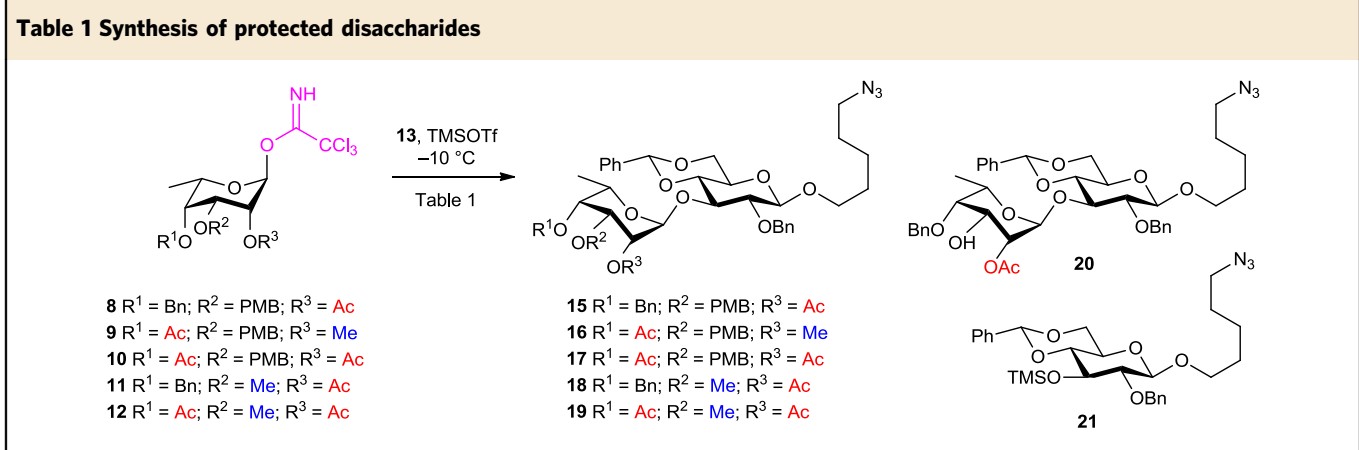

| | | | | | | |
|---|---|---|---|---|---|---|
| **8** R¹ = Bn; R² = PMB; R³ = Ac | | | **15** R¹ = Bn; R² = PMB; R³ = Ac | | | |
| **9** R¹ = Ac; R² = PMB; R³ = Me | | | **16** R¹ = Ac; R² = PMB; R³ = Me | | | |
| **10** R¹ = Ac; R² = PMB; R³ = Ac | | | **17** R¹ = Ac; R² = PMB; R³ = Ac | | | |
| **11** R¹ = Bn; R² = Me; R³ = Ac | | | **18** R¹ = Bn; R² = Me; R³ = Ac | | | |
| **12** R¹ = Ac; R² = Me; R³ = Ac | | | **19** R¹ = Ac; R² = Me; R³ = Ac | | | |

| Entry | Donor (equivalents) | Solvent[a] | 4 Å MS[b]/time (h) | TMSOTf (equivalents) | Product yield (%)[c] | Ratio α/β[d] |
|---|---|---|---|---|---|---|
| 1 | **8** (1.3) | DCE | +/21 | 0.2 | **15** (30)[e] | α only |
| 2 | **8** (2.0) | Et₂O | +/1 | 0.2 | **15** (43)[f] | α only |
| 3 | **8** (1.5) | Et₂O | +/8 | 0.2 | **15** (78) | α only |
| 4 | **8** (2.0) | Et₂O | −/0.2 | 0.02 | **15** (95) | α only |
| 5 | **9** (2.0) | DCE | +/0.2 | 0.2 | **16** (51) | α only |
| 6 | **9** (2.0) | Et₂O | −/0.2 | 0.01 | **16** (90) | α only |
| 7 | **10** (2.0) | DCE | +/0.2 | 0.2 | **17** (44) | α only |
| 8 | **10** (2.0) | Et₂O/DCE 5:1[g] | −/0.2 | 0.01 | **17** (58) | α only |
| 9 | **11** (2.0) | Et₂O | −/0.2 | 0.01 | **18** (81) | α only |
| 10 | **12** (2.0) | Et₂O | −/0.2 | 0.01 | **19** (76) | α only |

*DCE* 1,2-dichloroethane, *Et₂O* diethyl ether, *MS* molecular sieves, *TMSOTf* trimethylsilyltrifluoromethanesulfonate
[a]Anhydrous solvent over molecular sieves (-0.05 M)
[b]With (+) or without (−) freshly activated powdered molecular sieves
[c]Isolated yield
[d]Determined by ¹H NMR
[e]Disaccharide **20** was isolated as the major compound
[f]Silylated derivative **21** was isolated in 42% yield
[g]DCE was added to ensure the solubility of donor

Switching DCE for diethyl ether (Et₂O) as the solvent slightly increased the yield of disaccharide **15** (from 30% to 43%) while preventing the formation of disaccharide **20**; however, silylated glucose derivative **21** was isolated as a by-product. Increasing the reaction time from 1 h (entry 2) to 2 h (entry 3) enabled the conversion of silylated derivative **21** into disaccharide **15**, thereby enhancing the yield to 78%. We then discovered that performing the glycosylation without MS had a dramatic effect on the reaction kinetic and yield. Under these conditions (entry 4), reaction time was shortened to 20 min, only 0.02 equivalent of TMSOTf was needed, and the yield went up to 95% without PMB deprotection. The other disaccharides (**16–19**) were conveniently synthesized using these optimized conditions (entries 6, 8–10). Pleasingly, the glycosylation reactions were fully α-stereoselective for all disaccharides, even without participating group at C2, such as for 2-O-methylated donor **9**, and the anomeric configuration was ascertained by undecoupled ¹³C NMR ($^1J_{C1,H1}$ = 174–176 Hz).

**Synthesis of protected trisaccharides**. With disaccharides **15–17** in hand, we then turn our attention to the synthesis of trisaccharides **24–26** (Table 2). Cleavage of the PMB group was performed under the action of 2,3-dichloro-5,6-dicyano-1,4-benzoquinone (DDQ) in dichloromethane (DCM) at room temperature affording disaccharides **20**, **22**, and **23** in very good yields (77–87%) and, importantly, without noticeable acetyl migration to the C3 position. Glycosylation of disaccharide **22** with thioglucoside **14** (entry 1) under the combined action of N-iodosuccinimide (NIS) and silver(I) trifluoromethanesulfonate (AgOTf)[51] at −10 °C in an Et₂O/DCE mixture led to trisaccharide **25** in 65% yield as the sole β-anomer. Applying these conditions

to the synthesis of trisaccharide **26** also gave rewarding results (entry 2). However, we were surprised to find that glycosylation of disaccharide **20**, bearing a Bn group at C4, was not successful under these conditions (entry 3); instead degradation of donor was revealed by thin layer chromatography. We then tested several glycosylation conditions (the most relevant are shown in entries 4–10) using disaccharide **20** as an acceptor but without any success, as only traces of trisaccharide **24** were detected. When the reaction was performed in DCM at −78 °C using NIS/AgOTf as the promoter, dimerization of donor **14**, yielding diglucoside **29**, was observed (entry 5). Activation of thioglycoside **14** under the action of CuBr₂ in the presence of tetrabutylammonium bromide[52] was attempted in order to generate a more reactive bromide species[53]. However, this reaction mainly led to disaccharide **27** in which the acetyl group had migrated from the C2 to the C3 position (entry 6). Anomeric fluoride **S26** (entry 8) as well as N-phenyl-2,2,2-tri-fluoroacetimidate **S27** (entries 9 and 10) were also evaluated as donors but both failed to provide trisaccharide **24**.

We hypothesized that the steric hindrance and electronic effect of the Bn group at C4 can be invoked to explain these negative results. Therefore, 6-deoxytalose building block **S30** bearing a less hindered, electron-withdrawing Lev group at C4 together with a chloroacetyl (ClAc) group at C3 was prepared (Supplementary Fig. 6). Unfortunately, we were not able to selectively deprotect the ClAc group under a variety of conditions and therefore this route was abandoned. Regioselective glycosylation of diol **S29** bearing a Lev group at C4 was also investigated (Supplementary Fig. 7). Using thioglycoside **14** under the promotion of dimethyl(methylthio)sulfonium trifluoromethane-sulfonate (DMTST)[54] in the presence of 2,6-di-*tert*-butyl-4-

**Table 2 Synthesis of protected trisaccharides**

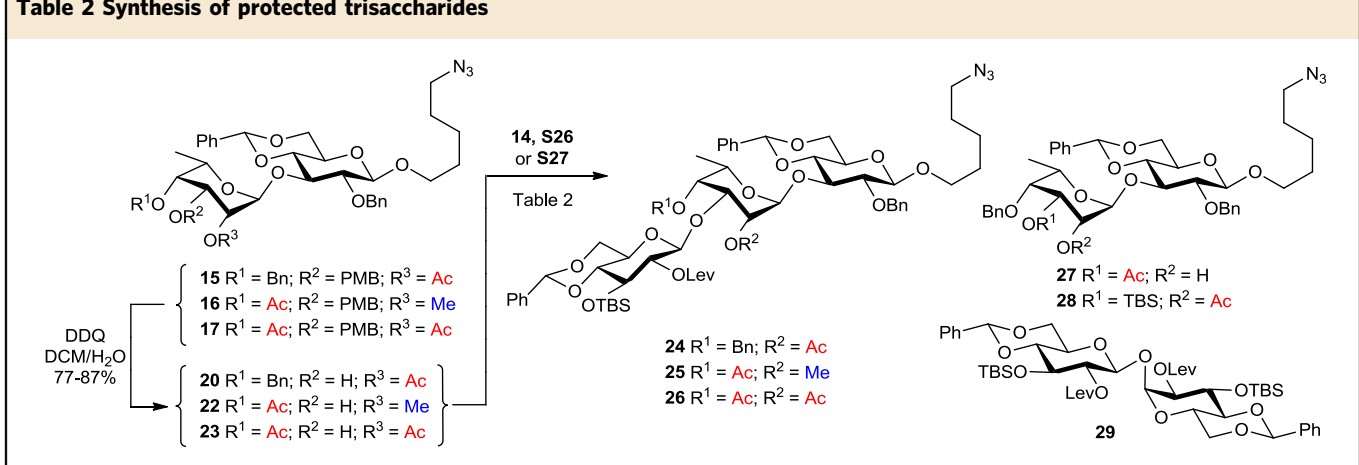

| Entry | Acceptor | Donor[a] | Promoter[b] | Solvent[c] | T (°C) (time (h)) | Product | Yield (%)[d] |
|---|---|---|---|---|---|---|---|
| 1 | 22 | 14 | NIS, AgOTf | Et$_2$O/DCE | −10 (0.2) | 25 | 65[e] |
| 2 | 23 | 14 | NIS, AgOTf | Et$_2$O/DCE | −10 (0.2) | 26 | 50[e] |
| 3 | 20 | 14 | NIS, AgOTf | Et$_2$O/DCE | −10 (0.2) | 24 | ND[f] |
| 4 | 20 | 14 | NIS, AgOTf | DCE | −10 (1) | 24 | ND[f] |
| 5 | 20 | 14 | NIS, AgOTf | DCM | −78 (3) | 24 | ND[g] |
| 6 | 20 | 14 | CuBr$_2$, TBAB | DCM/DMF | 22 (72) | 27 | 90 |
| 7 | 20 | 14 | DMTST, DTBMP | DCE | 40 (48) | 24 | ND[h] |
| 8 | 20 | S26 (F) | SnCl$_2$, AgOTf | Et$_2$O/DCM | −10 (0.3) | 24 | ND[f] |
| 9 | 20 | S27 (PTFA) | TMSOTf | DCE | −10 (0.2) | 24 | ND[f] |
| 10[i] | 20 | S27 (PTFA) | TBSOTf | tol | 75 (2) | 28 | 60 |

*AgOTf silver(I) trifluoromethanesulfonate, DCM dichloromethane, DDQ 2,3-dichloro-5,6-dicyano-1,4-benzoquinone, DMF N,N-dimethylformamide, DMTST dimethyl(methylthio)sulfonium trifluoromethanesulfonate, DTBMP 2,6-di-tert-butyl-4-methylpyridine, NIS N-iodosuccinimide, TBAB tetrabutylammonium bromide, TBSOTf tert-butyldimethylsilyl trifluoromethanesulfonate, tol toluene*
*[a]Donor was used in excess (1.5 equivalents)*
*[b]The reaction was performed adding freshly activated powdered molecular sieves*
*[c]Anhydrous solvent over molecular sieves (~0.05 M)*
*[d]Isolated yield*
*[e]Only the β-anomer was detected by ¹H NMR*
*[f]Degradation of donor*
*[g]The dimer 29 was detected as the major compound*
*[h]No reaction*
*[i]Inverse procedure*

methylpyridine led to the formation of disaccharide S32, having unfortunately the wrong regioselectivity.

In an attempt to enhance the regioselectivity of the glucosylation reaction at C3, 2-aminoethyl diphenylborinate catalyst was used following the conditions recently developed by Taylor and colleagues[55] (Supplementary Fig. 8). Thus triol S3 was reacted with perbenzylated glucose chloride S33 in the presence of Ag$_2$O in acetonitrile with a catalytic amount of 2-aminoethyl diphenylborinate. A mixture of regioisomeric disaccharides was formed in which desired disaccharide S34 was isolated in 25% yield. The poor glycosylation yield coupled with anticipated difficulties for the subsequent selective monoacetylation at either C2 or C4 position led us to consider using diol S5 instead. However, the presence of a Bn group at C4 reversed the regioselectivity of the glycosylation under Taylor conditions giving disaccharide S35 in 58% yield following acetylation. At this point, it became obvious that the presence of protecting groups other than acetyl at the C4 position of the 6-deoxytalose residue hamper the glycosylation on the adjacent *cis* alcohol. Other synthetic avenues were thus investigated.

**Second-generation synthesis of protected trisaccharides**. On the basis of these previous results, we devised an alternative synthetic route in which an epimeric rhamnose moiety was glucosylated prior to its conversion into the talo-configuration. It was anticipated that the steric hindrance at the C4 position

would be avoided in such a case. Therefore, as depicted in Fig. 3, alcohol S1 was levulinoylated at C4 and the isopropylidene cleaved under acidic conditions to give diol 31 in 81% yield over two steps. Glucosylation using Taylor catalyst in the presence of 4 Å MS cleanly provided disaccharide 32 following acetylation (73% over two steps). The other regioisomer was not detected. Then the Lev group was removed using hydrazine acetate, the resulting alcohol oxidized with Dess–Martin periodinane in refluxing DCE[56], and the ketone reduced in the presence of NaBH$_4$ with full control of diastereoselectivity[57]. Attempts were made to protect the axial C4 position with a Lev group, yet, even under drastic conditions, only small amounts of the levulinoylated derivative were formed. We thus decided to go further leaving this hydroxyl free. Disaccharide 34 was transformed into the TCA derivative 35 in 65% over three steps involving: (1) isomerization of the allyl group using an iridium-based catalyst; (2) iodine-promoted hydrolysis; and (3) activation of the resulting hemiacetal into a TCA derivative. Then TCA 35 was coupled with glucose acceptor 13 in the presence of TMSOTf in an attempt to form a trisaccharide. However, acceptor 13 did not react while disaccharide 35 underwent rearrangement into tricyclic orthoester 36, which was unexpectedly stable[58, 59]. The "all-*cis*" conformation of this intriguing compound was confirmed by single-crystal X-ray diffraction (CCDC 1520384, Supplementary Tables 2, 3, 4, and 5). It is likely that orthoester 36 would come from the intramolecular attack of the free C4 alcohol on the dioxalenium ion

**Fig. 3** Second-generation synthesis of protected trisaccharides. Reagents and conditions: a Lev₂O, py, DMAP, 50 °C, 2 h, 99%; b 80% aq. HOAc, 60 °C, 6 h, 82%; c chloride donor **S33**, 2-aminoethyl diphenylborinate (0.25 equivalent), Ag₂O, CH₃CN, 4 Å MS, 60 °C, overnight, 74%; d Ac₂O, py, DMAP, RT, 3–4 h, 98% (for **32**); 94% (for **41**); e H₂NNH₂.HOAc, DCM, MeOH, RT, overnight 82%; f Dess–Martin periodinane, DCE, 70 °C, 1 h; g NaBH₄, MeOH/DCM 5:1, −10 °C to RT, 71% (for **34**, over two steps); 85% (for **39**, over two steps); h [Ir(COD){PMe(C₆H₅)₂}₂]⁺.PF₆⁻, H₂, THF, RT, 1 h; i I₂, THF, H₂O, RT, 2 h; j CCl₃CN, Cs₂CO₃, DCM, Me₂CO, RT, 2 h, 65% (for **35**, over three steps); 81% (for **37**, over three steps); k acceptor **13**, TMSOTf, 4 Å MS (only for **38**), Et₂O/DCE 5:1, −10 °C, 10 min, 41% (for **36**); l H₂NNH₂.H₂O, py, HOAc, 0 °C to RT, overnight, 77% (over two steps); m PDCP, DMSO, Et₃N, DCM, −10 °C to RT, 1 h; n NaOMe, MeOH/DCM 2:1, RT, overnight, 81%. Ac₂O acetic anhydride, CCl₃CN trichloroacetonitrile, COD cyclooctadienyl, DMAP 4-(dimethylamino)pyridine, DMSO dimethylsulfoxide, Et₃N triethylamine, HOAc acetic acid, Lev₂O levulinic anhydride, PDCP phenyl dichlorophosphate, py pyridine, RT room temperature, THF tetrahydrofuran

(Supplementary Fig. 9). Attempts to glucosylate compound **36** in the presence of TMSOTf in either DCE or Et₂O only led to orthoester degradation[60, 61].

In an ultimate synthetic sequence, conversion of the rhamno- into the talo-configuration was then attempted at the trisaccharide level (Fig. 3). Disaccharide **32** was converted into TCA derivative **37**, which was successfully coupled with acceptor **13** using the previously optimized conditions. The Lev group was cleaved under the action of hydrazine monohydrate to give trisaccharide **38** in 77% yield over two steps from TCA **37**. Oxidation of the free alcohol at C4 was performed using Dess–Martin periodinane, but degradation occurred and trisaccharide **39** was isolated in low yield following NaBH₄ reduction (31%, over two steps). By contrast, Pfitzner–Moffatt oxidation[62] of trisaccharide **38** using phenyl dichlorophosphate followed by subsequent reduction of the crude ketone cleanly provided target trisaccharide **39** in very good yield (85%, over two steps). The latter was deacetylated or acetylated under standard conditions to give trisaccharides **40** and **41**, respectively. A similar synthetic approach was successfully applied to the

second-generation synthesis of terminal disaccharides **6** and **7** (Supplementary Fig. 10).

**Deprotection of oligosaccharides**. The last step in the synthesis of target oligosaccharides **1**–**7** was the global deprotection of trisaccharides **25**, **39**, **40**, and **41** as well as disaccharides **18** and **19** (Fig. 4). In order to provide trisaccharide **4** bearing an acetyl group at C4, a three-step synthetic sequence was performed starting from protected trisaccharide **25**, which consisted in delevulinoylation using hydrazine acetate, cleavage of the TBS group by treatment with triethylamine trihydrofluoride in refluxing tetrahydrofuran (THF), and hydrogenolysis with Pearlman catalyst through microfluidic conditions (H-Cube) in the presence of HCl (2 equivalents). Under these conditions, monoacetylated trisaccharide **4** was obtained in 72% yield over three steps. Zemplén deacylation of trisaccharide **25**, cleavage of the TBS group using tetrabutylammonium bromide in THF followed by microfluidic hydrogenolysis led to non-acetylated trisaccharide **3** in 69% yield over three steps. Finally, deprotection of oligosaccharides **39**, **40**, **41**, **18**, and **19** was best performed

**Fig. 4** Global deprotection allowing access to target oligosaccharides. Reagents and conditions: a H$_2$NNH$_2$.HOAc, MeOH/DCM 5:2, RT, overnight; b TREAT-HF, THF, reflux, 24 h, 92% (over two steps); c H-Cube, 20% Pd(OH)$_2$/C, HCl (2.0 equivalents), 10 bars, 40 °C, MeOH/DCE, 78% (for **4**); 78% (for **3**); d NaOMe, MeOH/DCM 2:1, RT, overnight; e TBAF, THF, 0 °C to RT, overnight, 89% (over two steps); f Pd black, H$_2$, HCl (1.0 equivalent), MeOH/DCE, quant. (for **1**, **2**, **5**, **6**, and **7**). *TBAF* tetrabutylammonium fluoride, *TREAT-HF* triethylaminetrihydrofluoride

through heterogeneous hydrogenolysis conditions using Pd black and 1.0 equivalent of HCl in a DCE/MeOH mixture affording target oligosaccharides **2**, **1**, **5**, **6**, and **7**, respectively, in quantitative yields. Importantly, using excess of HCl partially cleaved the acetyl group at C2, which was found to be more labile than the one at C4.

**Reactivity of the oligosaccharides with LPS-specific mAbs.** Several previous studies have identified mAbs that differentially recognize *Bp* or *Bm* LPS antigens[23, 26, 43, 47, 63]. Notably, mAb Pp-PS-W is specific for *Bp* OAg while mAbs 4C7, 3D11, and 9C1-2 are specific for *Bm* OAg[43, 47]. Although the OAgs expressed by *Bp* and *Bm* are structurally similar, *Bm* OAg lacks 4-*O*-acetyl substitutions on talose residues, a key difference that influences recognition of these antigens by mAbs[43, 47]. The structures of *Bp* (RR2808) and *Bm*-like (RR4744) OAgs and their corresponding mAb reactivity profiles are shown in Fig. 5a and Supplementary Fig. 167. To determine whether the oligosaccharides synthesized in this study were recognized by the various LPS-specific mAbs, ELISAs were conducted using all seven oligosaccharides along with LPS controls. Results demonstrated that mAbs 4C7, 3D11, and 9C1-2 reacted strongly with disaccharide **6**, which represents the capping residue associated with *Bm* OAg, and that mAb Pp-PS-W reacted strongly with disaccharide **7**, which represents the capping residue associated with *Bp* OAg (Fig. 5b). These findings are consistent with the LPS reactivity patterns observed and indicate that all of the mAbs tested appear to recognize the terminal residues of the either *Bp* or *Bm* OAgs. Additionally, these results confirm our previous work showing that mAb Pp-PS-W reacts only with →3)-β-D-glucopyranose-(1→3)-6-deoxy-α-L-talopyranose-(1→ polymers in which the 6-deoxytalose residues are coordinately acetylated at the *O*-2 and *O*-4 positions[43]. Importantly, as mAbs Pp-PS-W, 4C7, and 9C1-2 have been shown to be passively protective in animal models of melioidosis

or glanders, our data support the use of disaccharides **6** and **7** as components of novel vaccine candidates.

**Kinetic characterization of mAb 4C7/oligosaccharide interactions by SPR.** SPR[64] was used for a real-time analysis of the binding affinities between mAb 4C7 and the synthetic oligosaccharides (Fig. 6 and Supplementary Figs. 168 and 169). mAb 4C7 was selected as a model IgG as it has recently been shown to provide significant protection of mice from a lethal challenge with *Bp* in the course of a passive immunization protocol[28]. Disaccharide **6**, which presented the highest recognition toward mAb 4C7 in the ELISA assay, was evaluated by SPR as well as disaccharide **7**, and trisaccharide **2**, which features the major intrachain epitope of *Bp*/*Bm* OAg. Therefore, oligosaccharides **2**, **6**, and **7** were biotinylated using NHS ester chemistry and the resulting constructs (**BIO-2**, **BIO-6**, and **BIO-7**, respectively, Fig. 6a) were immobilized on the surface of a streptavidin (SA)-coated sensor chip (Supplementary Figs. 11 and 170). Different concentrations of mAb 4C7 were injected for 180 s, followed by passive dissociation for 300 s. The changes in refractive index at the sensor chip surface, which reflect the magnitude of the interactions, were monitored and recorded in arbitrary response units. The kinetics of binding between mAb 4C7 and the biotinylated oligosaccharides were illustrated in the sensorgrams, which are plots of response units vs time. According to the sensorgrams, mAb 4C7 bound to immobilized **BIO-6** and **BIO-7**, but did not interact with immobilized **BIO-2** (Fig. 6b). The $K_D$ values, which were calculated using a steady-state affinity model, demonstrated that mAb 4C7 had a higher affinity binding to **BIO-6** (22 nM) as compared with **BIO-7** (120 nM). In agreement with the results obtained by ELISA, the SPR-binding results indicate that mAb 4C7 tightly interacts with the terminal methylated talose residue found at the non-reducing end of *Bm*-like LPS OAg. Furthermore, the presence of an acetyl

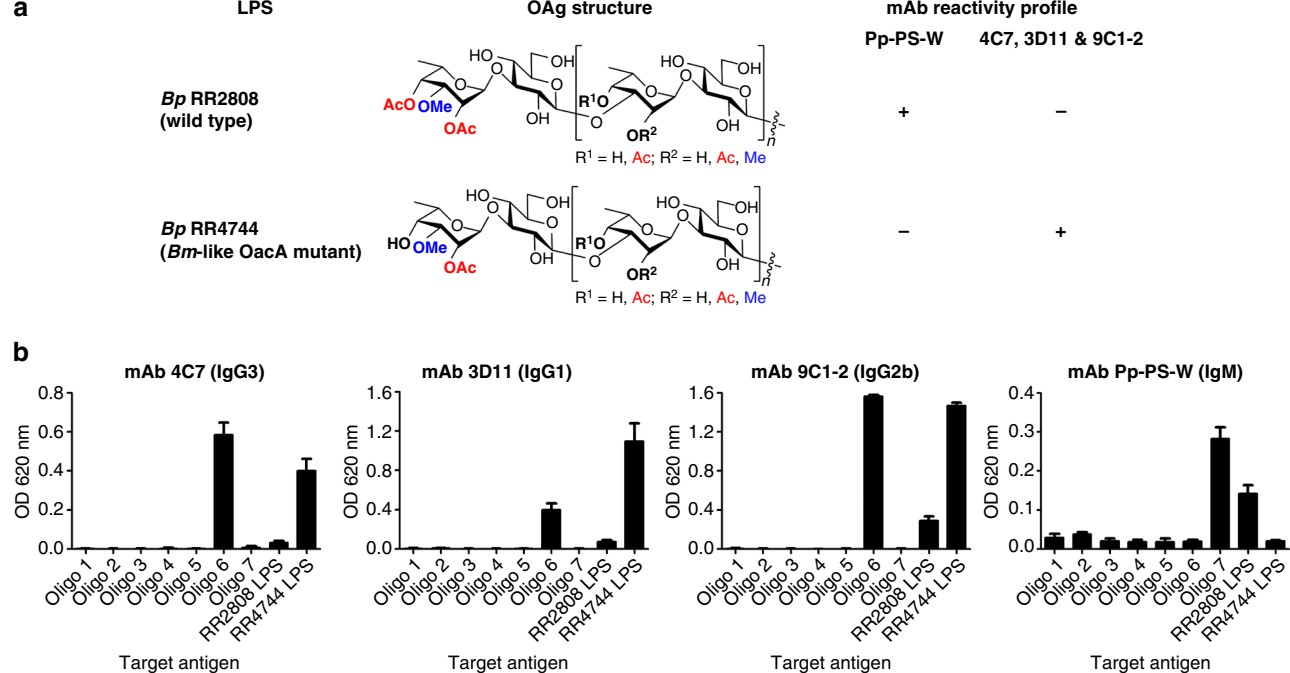

**Fig. 5** Interactions of LPS-specific mAbs with synthetic oligosaccharides. **a** Reactivity profiles of mAbs Pp-PS-W, 4C7, 3D11, and 9C1-2 with LPS antigens purified from *Bp* strains RR2808 and RR4744 (see Supplementary Fig. 167). **b** Reactivity profiles of the mAbs with synthetic oligosaccharides **1**–**7**, RR2808, and RR4744 LPS as determined by ELISA. *Black bars* represent the mean±sd of assays conducted in triplicate

group at the C4 position of the talose unit significantly hampers the binding with mAb 4C7 by a five-fold order of magnitude.

**Binding epitopes of mAb 4C7 with oligosaccharides by STD-NMR.** In order to dissect, at a molecular level, the binding of mAb 4C7 to *Bp* and *Bm* OAg, we employed *ad hoc* NMR techniques aimed to identify and characterize the interactions of synthetic oligosaccharides to the monoclonal antibody[65]. STD-NMR spectroscopy is well suited to derive deep insights on the molecular features that govern antigen recognition from antibodies characterized by weak or medium affinity[66]. STD-NMR experiments were carried out with disaccharides **6** and **7** that differ in the acetylation pattern at the C4 position of the talose residue. The STD-NMR spectra performed on the mAb–disaccharide **7** mixture at 298 K did not show any signals (Supplementary Fig. 171) likely due to unfavorable binding kinetics. As the temperature strongly influences the kinetics and consequently the observed STD effects[67], we ran STD-NMR spectra at different temperatures (Supplementary Fig. 172 and Fig. 7b). Interestingly, at 283 K, some STD enhancements were observed for the mAb 4C7–disaccharide **7** complex (Fig. 7b). However, the characterization of the ligand epitope mapping of disaccharide **7** was hampered as only very low STD-NMR effects were observed. STD-NMR measurements gathered on disaccharide **6**, instead, allowed deducing a more accurate binding epitope, detecting the ligand region in closer contact to the antibody. A qualitative analysis of STD enhancements clearly evidenced the involvement of both glucose and talose moieties, which were both recognized by the mAb 4C7 (Fig. 7a). However, the strongest STD effects all belonged to the terminal talose unit, with the proton at position 2 experiencing the highest transfer of saturation (100% normalized STD effect). In addition, *O*-acetyl group (97%), H1 (96%), and H4 (88%) exhibited large STD enhancements indicating that they were important as well for antibody binding. Less pronounced STD signals were observed for protons of the glucose residue revealing that they participated

to a minor extent in the interaction with mAb 4C7. In detail, proton H3 showed an STD effect close to 60%, whereas protons at positions 4, 5, and 6 displayed even lower STD intensities (<50%). Therefore, STD-NMR data suggest that the main contact surface area was positioned within the talose residue thus highlighting its role in the binding process, whereas the glucose moiety less contributed to the interaction with the antibody. In addition, considering the high contribution to the binding of hindered proton H4 in disaccharide **6**, this could explain why the presence of an acetyl group at this position, such as for disaccharide **7**, significantly weakens the binding with mAb 4C7 resulting in slight STD effects.

**Immunization of mice with disaccharide- and OAg-based glycoconjugates.** Extending upon the observation that disaccharides **6** and **7** reacted with *Bm* and *Bp* LPS-specific mAbs, respectively, we next wanted to determine whether these synthetic oligosaccharides were capable of stimulating immune responses in mice. Using NHS ester chemistry, disaccharides **6** and **7** were individually coupled to CRM197 resulting in the semi-synthetic oligosaccharide conjugates **SOC-6** and **SOC-7** (Supplementary Fig. 12). Following conjugation, the samples were examined by sodium dodecyl sulfate-polyacrylamide gel electrophoresis (SDS-PAGE). Results of these analyses demonstrated that, in both instances, the disaccharides had covalently linked to the protein carrier, as indicated by the shifts in molecular weights of the glycoconjugates relative to the molecular weight of the unconjugated CRM197 control (Supplementary Fig. 173). Additionally, western immunoblotting confirmed that the structural integrity/antigenicity of the disaccharide moieties remained intact following coupling to the protein carrier based upon their reactivity with mAbs 4C7, 3D11, and 9C1-2 or Pp-PS-W (Supplementary Fig. 173). Further analysis of the constructs by matrix assisted laser desorption/ionization time-of-flight mass spectrometry (MALDI-TOF-MS) revealed that **SOC-6** and **SOC-7** consisted of about six and five disaccharides

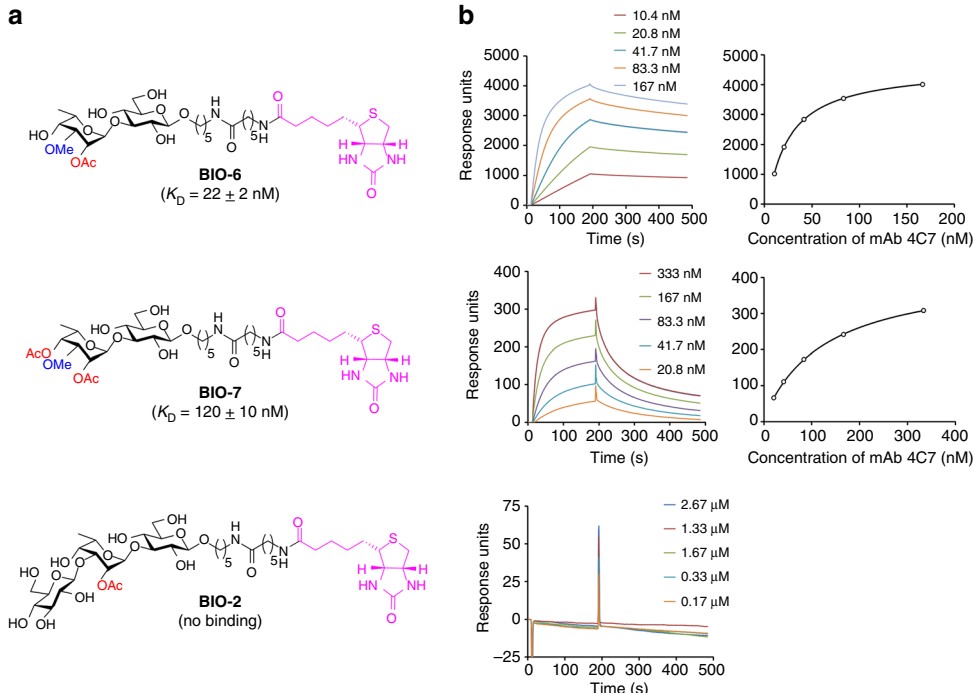

**Fig. 6** $K_D$ values of mAb 4C7 binding to biotinylated oligosaccharides inferred by SPR. **a** Chemical structures of the biotinylated oligosaccharides with their corresponding $K_D$ values. The compounds were immobilized on the surface of a streptavidin-coated sensor chip. Samples (two-fold serial dilution of mAb 4C7) were injected over the sensor surface for 180 s (association), after which the mAb was allowed to passively dissociate for 300 s. $K_D$ values were calculated with a steady-state affinity model (response units vs concentration plots). Indicated $K_D$ values are the mean±sd of three runs. **b** Representative sensorgrams and steady-state affinity model fitting for each corresponding biotinylated oligosaccharides. See Supplementary Methods and Supplementary Figs. 11 and 168–170 for details

covalently linked to CRM197, respectively (Supplementary Fig. 174). The conjugates were ~95% protein (w/w) as measured by BCA assay.

To examine the immunogenic potential of the disaccharide-based glycoconjugates, groups of BALB/c mice were immunized with **SOC-6** or **SOC-7**. ELISAs were used to assess the reactivity of the immune serum samples with disaccharides and OAgs (Fig. 8a) while immunofluorescence microscopy was used to assess reactivity with whole cells (Supplementary Fig. 175). Results showed that **SOC-6** stimulated significantly higher antigen-specific IgG titers than did **SOC-7**. For disaccharide-specific responses, the end point titers elicited by **SOC-6** ranged from 1:400 to 1:64,000 while the end point titers achieved for **SOC-7** ranged from 1:200 to 1:800. Similar trends were observed for OAg-specific responses with **SOC-6** end point titers (vs RR4744 OAg) ranging from 1:400 to 1:64,000 and **SOC-7** end point titers (vs RR2808 OAg) ranging from 0 to 1:200. For control purposes, BALB/c mice were immunized with the OAg-based glycoconjugates OC-4744 and OC-2808. Consistent with the results shown in Fig. 8a, OC-4744-immunized mice demonstrated high-titer IgG responses against both disaccharide **6** and RR4744 OAg with end point titers ranging from 1:800 to 1:409,600 and from 1:128,000 to 1:512,000, respectively (Fig. 8b). In contrast, mice immunized with OC-2808 exhibited high-titer responses against RR2808 OAg (1:32,000–1:256,000) but failed to produce strong responses against disaccharide **7** (1:200–1:800). Similar results were also obtained when C57BL/6 mice were immunized with OC-2808 (Supplementary Fig. 176).

**Human immune responses to Bp OAg.** Based on our mouse studies, high-titer antibody responses that recognize the terminal disaccharide of *Bm* OAg could be produced by immunization with either **SOC-6** or OC-4744. In contrast, high-titer antibody responses that recognize the terminal epitope of *Bp* OAg could not be raised by immunization with either **SOC-7** or OC-2808. Potential reasons for this might be that the 4-O-acetyl group on the capping residue has a role in modulating immune responses against the *Bp* OAg or that mice have a hole in their B-cell repertoire against this motif. To investigate this, ELISAs were used to assess the reactivity of culture-confirmed Thai melioidosis patient and Thai healthy donor serum samples with RR2808 OAg and disaccharide **7**. As shown in Fig. 8c, immune serum samples exhibiting reactivity with *Bp* OAg also had the capacity to crossreact with disaccharide **7**. These results indicate that, unlike mice, humans have the ability to generate antibody responses against the terminal disaccharide of *Bp* OAg. Collectively, our findings suggest that the inability of mice to raise antibodies against the terminal epitope of *Bp* OAg may be a species-restricted phenomenon. Additional studies will be required to further investigate this observation as well as identify alternative animal models to help overcome this issue.

**Discussion**

In this study, we have been successful in synthesizing a unique series of oligosaccharides featuring all of the intrachain and terminal epitopes found within the LPS OAgs from *Bp* and *Bm*. The optimal approach involved the epimerization of the C4 position of a 3-O-methylated or 3-O-glucosylated L-rhamnose building block at a late stage of the synthetic route, generating terminal disaccharides **6** and **7**, and intrachain trisaccharides **1–5**, respectively. All of the glycosylation reactions were fully stereoselective, the coupling products were obtained in high yields, and, importantly, no acetyl migration was detected at any steps of the synthetic sequence. The knowledge learned from

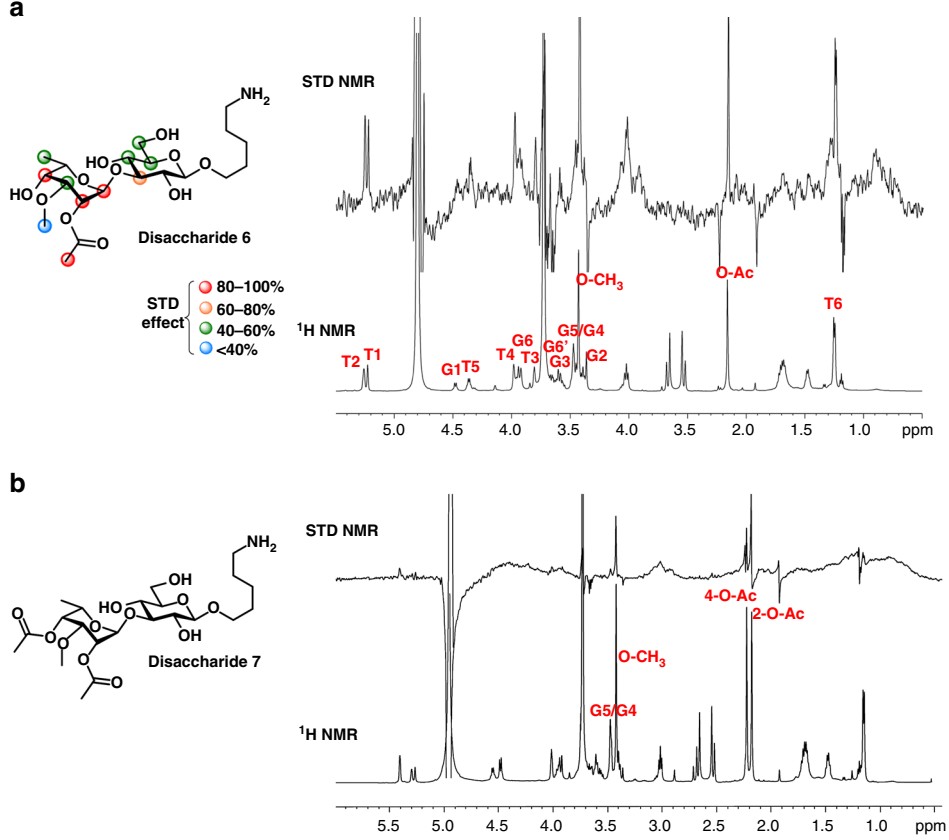

**Fig. 7** Epitope mapping of disaccharides/mAb 4C7 interactions probed by STD-NMR. Chemical structures and epitope binding of disaccharides **6a** and **7b** to mAb 4C7 along with reference [1]H and STD NMR spectra at 298 and 283 K, respectively. *Color code* indicates the percentages of STD effects. Low and not quantifiable STD effects were detected for disaccharide **7**. Both STD 1D NMR spectra were run with a 1:100 mAb 4C7/disaccharide mixture. The irradiation frequency was set at 8 ppm and a saturation time of 2 s was used. The proton resonances belonging to talose and glucose residues were indicated with *letters*, **T** and **G**, respectively

this synthetic journey could be used as a base for the elaboration of longer oligosaccharide chains related to *Bp* and *Bm* OAg, which would feature, for instance, both intrachain and terminal epitopes.

The synthetic oligosaccharides were used to probe and characterize the minimal binding epitopes for a series of *Bp* and *Bm* LPS-specific mAbs, which have been shown to be passively protective in mouse models of melioidosis and glanders. To do so, biochemical and biophysical approaches, including ELISA assay, SPR, and STD-NMR, were employed to study the interactions of synthetic oligosaccharides **1**–**7** with various mAbs. The results of the ELISA assay strongly suggest that mAbs Pp-PS-W, 4C7, 3D11, and 9C1-2 are targeted to the terminal residues found at the non-reducing end of *Bp* and *Bm* OAgs. The interaction between mAb 4C7, which recognizes the *Bm*-like capping residue, and disaccharides **6** and **7** was further investigated by STD-NMR. These NMR analyses revealed that mAb 4C7 primarily binds to the 6-deoxy-L-talose residue of disaccharide **6**, especially with the *O*-acetyl group and protons at the C1, C2, and C4 positions which experienced the higher STD effects, and, to a lesser extent, with the glucose residue. In contrast, only weak STD effects were detected for disaccharide **7**, a result that could be explained by the presence of a supplemental acetyl group at the C4 position. SPR measurements with biotinylated disaccharides (**BIO-6** and **BIO-7**) in the presence of mAb 4C7 supported this behavior. Indeed, disaccharide **6** was shown to bind more strongly to mAb 4C7 than disaccharide **7**, with a $K_D$ value in the low nanomolar range.

These results prompted us to evaluate the immunogenicity of disaccharides **6** and **7** in mice. To generate the semisynthetic glycoconjugates **SOC-6** and **SOC-7**, disaccharides **6** and **7** were covalently linked to CRM197. Mice immunized with **SOC-6** produced high-titer IgG responses that were raised against the disaccharide component of the constructs. Importantly, these responses were crossreactive with *Bm*-like OAgs. Optimization of the loading level as well as the multivalent display[68] of disaccharide epitopes could help improve the immunogenicity of the constructs. Moreover, the straightforward and high-yielding synthesis of disaccharide **6** represents an asset for the industrial and cost-effective production of such vaccines. Thus **SOC-6** stands as a promising vaccine candidate to be tested in animal models of glanders.

In summary, our results highlight the importance of *O*-acetyl and *O*-methyl modifications for recognition of OAgs by *Bp* and *Bm* LPS-specific mAbs. Furthermore, our findings support the use of synthetic chemistry for deciphering the immunogenic epitopes of non-stoichiometrically substituted surface polysaccharides in the context of antibacterial glycoconjugate vaccines. Collectively, it is anticipated that these studies will serve as foundation for the development of novel therapeutics, diagnostics, and vaccine candidates to combat diseases caused by *Bp* and *Bm*.

## Methods

**Chemical synthesis**. The complete experimental details, compound characterization data, and X-ray crystallographic data can be found in

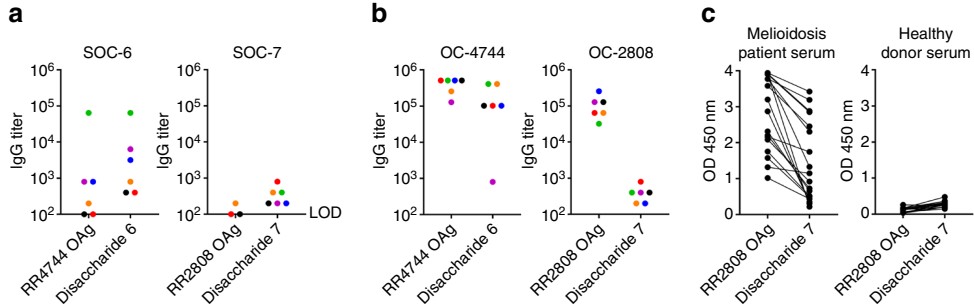

**Fig. 8** Mouse and human immune responses to disaccharides and OAgs. BALB/c mice ($n = 6$ per group) were immunized with **a** SOC-6 and SOC-7 or **b** OC-4744 and OC-2808. ELISAs were used to quantitate immune serum IgG titers. *Colored dots* represent the mean end point titers for individual mice against the various target antigens. LOD, limit of detection. **c** Serum samples from culture-confirmed Thai melioidosis patients ($n = 18$) and Thai healthy donors ($n = 18$) were assayed for reactivity with the target antigens using single-dilution ELISAs. *Connecting lines* indicate identical serum samples

Supplementary Methods. For the NMR spectra of new compounds, see Supplementary Figs. 13–166.

**mAb 4C7 production**. mAb 4C7 was produced as previously described[63]. Briefly, BALB/c mice were intraperitoneally injected with $2 \times 10^8$ CFU of heat-inactivated *Bp* (strain 1026b) every 2 weeks for an 8-week period. The antibody titers to *Bp* were monitored using an indirect ELISA with the heat-inactivated strain 1026b in the solid phase. The last immunization was administered 3 days prior to sple-nectomy. Splenic cells were fused with myeloma cells to produce mAb-secreting hybridomas as previously described[69]. Western blotting analysis was performed to identify hybridoma clones that were producing mAbs reactive in the typical ladder-banding pattern of LPS[63]; as a result, the clone 4C7 was identified. To produce mAb 4C7, the hybridoma cell line was grown in Integra CL 1000 culture flasks (Integra Biosciences), and the mAb was purified by protein A affinity column chromatography.

**ELISA assays**. To assess the reactivity of LPS-specific mAbs (Pp-PS-W, 4C7, 3D11, and 9C1-2) with synthetic oligosaccharides **1**–**7** and *Burkholderia* OAgs, maleic anhydride 96-well plates (Pierce) were coated overnight at 4 °C with oligo-saccharides **1**, **2**, **3**, **4**, **5**, **6** or **7** (5 µg/ml) or purified LPS (10 µg/ml) solubilized in carbonate buffer (pH 9.6). The LPS antigens used in this study were purified from *Bp* strains RR2808 ($\Delta wcbB$; *Bp* LPS) and RR4744 ($\Delta wcbB \Delta oacA$; *Bm*-like LPS) as previously described[32, 45]. The coated plates were blocked at room temperature for 30 min with StartingBlock T20 (TBS) Blocking Buffer (SB; Pierce) and then incubated for 1 h at 37 °C with the various mAbs diluted 1/2000 in Tris-buffered saline + 0.05% Tween 20 (TBS-T)+10% SB. To facilitate detection, the plates were incubated for 1 h at 37 °C with 1/2000 dilutions of goat anti-mouse IgM or IgG-horse radish peroxidase (HRP) conjugates (Southern Biotech). The plates were then developed with TMB substrate (KPL) and read at 620 nm. The data were plotted and analyzed using GraphPad Prism 5 (GraphPad Software Inc.).

**SPR experiments**. SPR analysis of binding between mAb 4C7 and synthetic oligosaccharides was performed using a Biacore X-100 instrument (GE Health-care). HBS-EP + buffer (10 mM HEPES, 150 mM NaCl, 3 mM EDTA, and 0.05% $v/v$ Surfactant P20, pH 7.4, GE Healthcare) was used as a running buffer and diluent throughout the experiments. Biotinylated oligosaccharides **BIO-6**, **BIO-7**, and **BIO-2** (see Supplementary Methods) were separately immobilized on the surface of a SA-coated sensor chip (GE Healthcare); a second flow cell surface was left unmodified for reference subtraction. To generate sensorgrams, two-fold serial dilutions of mAb 4C7 were injected over the sensor chip surface with a flow rate of 30 µL/min for 180 s, followed by passive dissociation for 300 s. Between each cycle, the chip surface was regenerated with a 60 s pulse of 20 mM NaOH. Each analysis was performed in triplicate. Binding affinities ($K_D$) were calculated using the steady-state affinity model in the BIA evaluation software (version 2.0.1, GE Healthcare).

**STD-NMR experiments**. NMR experiments were performed with a Bruker 600 MHz DRX instrument equipped with a cryo probe at 283, 298, and 310 K. All the samples were dissolved in deuterated phosphate buffer (pH 7.4) and spectra were calibrated with internal sodium [D$_4$](trimethylsilyl)propionate (10 µm) at 0.0 ppm for $^1$H NMR. The ligand resonances were assigned by using standard NMR experiments. Samples for STD-NMR contained an mAb/ligand molar ratio from 1:50 to 1:100 and the antibody concentration was 12 µM. STD-NMR experiments were carried out with 32k data points and zero filled to 64k data point prior processing. A total of 4000 scans were recorded. Selective on-resonance irradiation of antibody resonances was performed at 8 ppm; the off-resonance frequency was set at 100 ppm. The antibody saturation was achieved by using a pulse train of Gaussian shaped pulses of 50 ms duration and 1 ms interpulse delay with an

irradiation power of 50 Hz. The saturation time was set at 2 s and a relaxation delay of 4 s was used. A T1ρ filter (50 db spin-lock pulse) and water suppression using excitation sculpting were applied. STD-NMR spectra of ligands in the absence of the antibody and spectra with antibody alone were acquired to obtain reference experiments. The STD effects were calculated by $(I_0 - I_{sat})/I_0$, where $(I_0 - I_{sat})$ is the intensity of the signal in the STD-NMR spectrum and $I_0$ is the peak intensity of the unsaturated reference spectrum (off-resonance). The STD signal with the highest intensity was set to 100%, and others were normalized to this. Data acquisition and processing were performed with TOPSPIN 3.2 software.

**Preparation and characterization of glycoconjugates**. Disaccharides **6** and **7** (200 µl of 15 mg/ml stocks in anhydrous dimethylsulfoxide (DMSO)) were added dropwise to disuccinimidylglutarate (DSG; 400 µl; 62.5 mg/ml stock in anhydrous DMSO) with trimethylamine (20 µl) and stirred for 2 h at room temperature. Phosphate-buffered saline (PBS; 800 µl, pH 7.2) was then added and the unreacted DSG was extracted twice with equal volumes of chloroform. The aqueous phase was recovered and reacted with CRM197 (2 mg, Reagent Proteins) solubilized in PBS (2 mL) at room temperature for 18–24 h. The reaction product was dialyzed extensively against dH$_2$O and concentrated using a 10 K MWCO Vivaspin Column (VIVAproducts). Conjugates were visualized by SDS-PAGE (4–12% Bolt gels; Life Technologies). Protein concentration was determined by BCA Assay (Pierce). The conjugates were further analysed by MALDI-TOF-MS. The results were acquired on a TOF/TOF 5800 System (AB SCIEX) using a linear positive mode. To improve ionization, the conjugated samples were dried and reconstituted with 50 mM ammonium bicarbonate buffer. The conjugates were mixed with 2,4,6-trihy-droxyacetophenone, which was used as the matrix for MALDI analysis. The resulting data were externally calibrated using bovine serum albumin. The disaccharide-based conjugates were named **SOC-6** and **SOC-7**, respectively. Glycoconjugates OC-4744 (RR4744 OAg+CRM197) and OC-2808 (RR2808 OAg +CRM197) were synthesized essentially as previously described[33]. The OAgs were purified from *Bp* RR2808 and RR4744 LPS as previously described[32, 45].

**Immunogenicity evaluation**. Groups of 6–8-week-old female BALB/c mice (Charles River) were immunized subcutaneously on days 0, 21, and 35 with 5 µg of the disaccharide-CRM197 glycoconjugates **SOC-6** and **SOC-7** or 10 µg of the OAg-CRM197 glycoconjugates OC-4744 and OC-2808 formulated in saline plus Alhydrogel 2% (500 µg/mouse; Brenntag) and PolyI:C (30 µg/mouse; InvivoGen). Terminal bleeds were conducted 14 days after the third immunization for the assessment of antibody responses. Six mice per group were chosen to qualitatively assess the immunogenicity of glycoconjugates. Therefore, no randomization, blinding, or statistical analysis was required for comparing the antibody levels. All procedures involving mice were performed according to protocols approved by the University of South Alabama Institutional Animal Care and Use Committee and were conducted in strict accordance with the recommendations in the Guide for the Care and Use of Laboratory Animals of the National Institutes of Health.

Antibody responses directed against disaccharides **6** and **7** as well as crossreactive responses against the *Burkholderia* OAgs were assessed by ELISA essentially as described above. To quantitate disaccharide-specific responses, maleic anhydride 96-well plates were coated with disaccharides **6** or **7** (5 µg/ml) solubilized in carbonate buffer. To quantitate OAg-specific responses, 96-well Maxisorp plates (Nunc) were coated with purified *Bp* RR2808 or RR4744 OAgs (1 µg/ml) solubilized in carbonate buffer. The OAgs were purified from *Bp* RR2808 and RR4744 LPS as previously described[32, 45]. The coated plates were blocked and then incubated for 1 h at 37 °C with the mouse serum samples serially diluted in TBS-T+10% SB. The plates were then incubated for 1 h at 37 °C with 1/2000 dilutions of anti-mouse IgG-HRP conjugate and developed as described above. The reciprocals of the highest dilutions exhibiting optical densities of two times background were used to determine the end point titers for the individual mice.

**Human serum ELISAs.** Serum samples from culture-confirmed Thai melioidosis patients ($n = 18$) and Thai healthy donors ($n = 18$) were assayed for reactivity with RR2808 OAg and disaccharide **7** essentially as previously described[70]. Plates were coated with RR2808 OAg or disaccharide **7** as described above. Serum samples were assayed at a fixed dilution of 1/2000. The study was approved by the Ethics Committee of Faculty of Tropical Medicine, Mahidol University (approval number MUTM 2014-079-02). Written informed consent was obtained from all subjects.

**Data availability**. The data that support the findings of this study are available from the corresponding authors (P.J.B. or C.G.) upon reasonable request. The X-ray crystallographic data of compound **36** (CCDC 1520384, Supplementary Tables 2, 3, 4, and 5) are available in the Supplementary Information file.

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

## Acknowledgements

This work was supported by a grant from the Agence Nationale de la Recherche Programme Jeunes Chercheuses Jeunes Chercheurs (ANR-JCJC-12-JS07-0003-01), the Natural Sciences and Engineering Research Council of Canada (NSERC, RGPIN-2016-04950), and the Department of Energy-funded (DE-FG02-93ER-20097) Center for Plant and Microbial Complex Carbohydrates. N.C. is supported by the National Institute of Allergy and Infectious Diseases of the National Institutes of Health (U01AI115520). M.M. thanks the Wroclaw Center of Biotechnology for PDoc funding within the framework of the Leading National Research Center (KNOW). The authors acknowledge the Unité de Chimie des Biomolécules at Institut Pasteur, Paris for kindly sharing their H-Cube instrument and Rosemary Roberts for technical assistance.

## Author contributions

M.T.K., M.M., A.N., and C.G. synthesized the oligosaccharides. M.T.K. and C.G. designed the synthetic experiments. Y.B. and J.M. performed the crystallographic study of compound **36**. T.N. and D.P.A. performed the SPR experiments. R.M., A.S., and A.M. performed the STD-NMR experiments. P.J.B. synthesized and characterized the glycoconjugates. P.J.B., T.L.S., and M.N.B. performed the immunogenicity study. K.S. and N.C. performed the human serum ELISAs. M.T.K., M.M., T.N., R.M., M.N.B., P.J.B., and C.G. wrote the manuscript. All authors read and approved the manuscript.

## Additional information

**Competing interests:** The authors declare no competing financial interests.

