## [Peer Review File · Nature Communications]

Reviewers' comments:

Reviewer #2 (Remarks to the Author):

The authors report the chemical synthesis of five trisaccharides and two disaccharides representing all of the reported inner and terminal fragments of the unique O-antigen polysaccharide of the lipopolysaccharides (LPS) of *Burkholderia pseudomallei* (Bp) and *Burkholderia mallei* (Bm). Bp and Bm are etiologic agents of melioidosis and glanders, respectively. These oligosaccharides bear an aminopentanyl aglycon therefore can be easily derivatized. The synthetic oligosaccharide derivatives are then subjected to binding studies with a series of Bp and Bm LPS-specific mAbs, which have previously been shown to be passively protective in mouse models of melioidosis and glanders. Biochemical and biophysical approaches, including ELISA assay, SPR, and STD-NMR are used and lead to conclusion that the mAbs target to the terminal disaccharide residues, especially the terminal 2-O-acetyl-3-O-methyl-6-deoxy-L-talose residue in disaccharide 6. Finally, the authors show that mice immunized with the terminal disaccharides covalently linked to CRM197 are able to produce high titer antibody responses that cross-reacted with Bm-like OAg antigens. These studies shall serve well as a starting point for the development of therapeutics, diagnostics, or vaccine candidates to combat diseases caused by Bp and Bm. The experiments are well done and data well presented. I recommend publication of this interdisciplinary work, with each part being performed by experts in the field, on *Nat. Commun.*

Minor points:

(1) In the synthesis part, the authors might comment on the unusual conditions used for the deprotection of the anomeric allyl group.

(2) In the binding studies, both disaccharides 6 and 7 show effective binding with mAb 4C7 in the ELISA and SPR assay (although the binding of 6 is indeed stronger), however, there is completely no binding of 7 in the STD-NMR experiments. The authors might comment on this. Additionally, the STD-NMR experiments show the strong involvement of the terminal talose residue in the binding with mAb, the authors might comment on if the appending 2-O-acetyl group also involves.

(3) Page 7, chloroacetyl group should not be abbreviated as AcCl (that is acetyl chloride) but ClAc.

(4) Page S13, Supplementary Fig. 12, the linker should be five carbon long instead of four carbon.

Reviewer #3 (Remarks to the Author):

This is an interesting study and expands on previous studies that demonstrated the protective efficacy of selected monoclonals. In this manuscript the authors go on to describe the recognised epitopes. From a biological prospective it is interesting and would guide further research and worthy of publication and likely to impact future vaccine design.

The strategy to link the antigens to the CRM is standard.

The authors demonstrated that the antibody responses recognised the OAg antigens. I would have like to have seen antibody binding to the whole bacteria - more biological relevance not just the antigens.

Since *B. pseudomallei* is an intracellular pathogen I would have like to have seen some additional experiments and discussion on the cellular immune responses. Especially since the authors use PolyI:C adjuvant (with Alhydrogel) that would up-regulate innate immune responses through activation of TLR3 and subsequently up regulation of IL-12 and IFNs.

On the NMR and SPR data I make the following statements:

6 and 7 seem both to have high affinity according to SPR but only 6 binds in the NMR experiment. This doesn't match with the fact that the second OAc group of 7 might abolish binding. It's still

nanomolar.

I find it also intriguing that the only OAc-group of 6 at the talose moiety is completely subtracted and not interacting at all although the talose is strongest engaging residue. This could only happen if the OAc is completely solvent exposed.

The authors should run the STD of 7 at 283 K and 310 K to see if there is a temperature effect. It's plausible though that a second OAc group might abolish binding.

I don't understand how the authors determine 40% of the hydroxyl group at C3 of 6 ? (blue circle Figure 7). Did they not use D₂O? This aspect doesn't make sense.

What are the T₁/T₂/.... in Figure 7? Needs to be labelled in the structure.

All in all the paper presents a range of interesting data, but to be accepted in the journal additional studies would need to be performed.

Reviewer #4 (Remarks to the Author):

In their manuscript entitled "Deciphering minimal antigenic epitopes associated with *Burkholderia pseudomallei* and *Burkholderia mallei* lipopolysaccharide O-antigens", Kenfack et al. describe a synthetic approach to access glycan epitopes of the Bp and Bm LPS O-antigens. This is a highly interesting and relevant study that highlights the utility of synthetic carbohydrate chemistry to identify immunogenic glycan epitopes as candidates for novel glycoconjugate vaccines. The authors' approach combines synthetic carbohydrate chemistry with a detailed characterization of the recognition of the minimal glycan epitopes by monoclonal antibodies (mAbs). Finally, they perform immunization studies in mice to determine the immunogenicity of the minimal glycan epitopes. The data on the molecular interaction of the synthetic oligosaccharides with LPS-specific mAbs as determined by glycan array are convincing. The performed SPR and STD-NMR measurements to determine K_D values and binding epitopes, respectively, allow for an in-depth insight into the crucial role of the talose moiety in the glycan/mAb interaction. The manuscript is well written and of interest to a broad readership. While the strength of the manuscript is clearly the chemistry part and the biophysical characterization of the glycan/mAb interactions, its weakness is the murine immunization studies. In my opinion, the in vivo studies need to be extended and fine-tuned to render the manuscript acceptable for publication (see "Specific points"):

Specific points:

1.) Why was a steady-state affinity model applied to calculate the K_D values from the SPR data (p.12) given that a mAb was used as analyte (i.e. a bivalent analyte)?

2.) The authors state that "future work [should] include optimizing the loading of oligosaccharides onto carrier proteins, varying the dose of the glycoconjugates delivered and determining the most effective adjuvant system" (p.15, ll. 462). In my opinion, however, the present data do not (yet) convincingly show strong immunogenicity of the selected glycan candidates (and the glycoconjugates used for immunization respectively). A prime-boost immunization protocol was employed (three immunizations in total) and a combination of the adjuvants Alum and PolyI:C was used. Still, the obtained IgG titers were fairly low given a limit of detection (LOD) of 100 (as shown in Figure 8C). While IgG responses were generally low for immunization with SOC-7, SOC-6 elicited higher antibody titers that, however, varied among mice (low titers in two out of six mice). I suggest trying alternative immunization protocols and adjuvants to obtain more reliable data on immunogenicity. In addition, the number of immunized mice could be increased to obtain more precise results for the endpoint titers.

3.) Along the same lines: The authors determined cross-reactivity of the antibodies induced by immunization with SOC-6 and SOC-7 with purified Bp O-antigens (Figure 8C). While strong cross-reactivity was observed upon immunization with SOC-6 in at least three out of six immunized mice, barely cross-reactive IgG responses were observed upon immunization with SOC-7. Still, even for SOC-6, there were generally marked reductions in IgG titers against the purified Bp O-antigens compared to the immobilized disaccharides (up to a half order of magnitude). What is the reason? For instance, have the authors determined antibody responses against the linker?

4.) Finally, it is well-known that LPS-specific monoclonal antibodies are protective in infection with Burkholderia and protection can be transferred by passive immunization (e.g. Trevino et al., *Infect. Immun.* 2006, 74, 1958; AuCoin et al., *PLOS One* 7, e35386, and other studies correctly cited by the authors). Since the potential of LPS-based subunit vaccines against Burkholderia has previously been shown, I consider a challenge study necessary in which the authors address the protective capacity of the glycoconjugates. Demonstrating the protective potential of CRM glycoconjugates would markedly strengthen the impact of the manuscript and would justify the authors' claim that "these studies serve as foundation for the development of novel therapeutics, diagnostics and vaccine candidates to combat diseases caused by Bp and Bm" as stated in the abstract (p.1).

RESPONSES TO REFEREES

Reviewer #2 (Remarks to the Author):

The authors report the chemical synthesis of five trisaccharides and two disaccharides representing all of the reported inner and terminal fragments of the unique O-antigen polysaccharide of the lipopolysaccharides (LPS) of *Burkholderia pseudomallei* (Bp) and *Burkholderia mallei* (Bm). Bp and Bm are etiologic agents of melioidosis and glanders, respectively. These oligosaccharides bear an aminopentanyl aglycon therefore can be easily derivatized. The synthetic oligosaccharide derivatives are then subjected to binding studies with a series of Bp and Bm LPS-specific mAbs, which have previously been shown to be passively protective in mouse models of melioidosis and glanders. Biochemical and biophysical approaches, including ELISA assay, SPR, and STD-NMR are used and lead to conclusion that the mAbs target to the terminal disaccharide residues, especially the terminal 2-O-acetyl-3-O-methyl-6-deoxy-L-talose residue in disaccharide 6. Finally, the authors show that mice immunized with the terminal disaccharides covalently linked to CRM197 are able to produce high titer antibody responses that cross-reacted with Bm-like OAg antigens. These studies shall serve well as a starting point for the development of therapeutics, diagnostics, or vaccine candidates to combat diseases caused by Bp and Bm. The experiments are well done and data well presented. I recommend publication of this interdisciplinary work, with each part being performed by experts in the field, on Nat. Commun.

Response: We thank the reviewer for his/her positive comments regarding our work.

Minor points:

(1) In the synthesis part, the authors might comment on the unusual conditions used for the deprotection of the anomeric allyl group.

Response: We agree with the reviewer that the conditions for the deprotection of the allyl group are somewhat unusual, *i.e.* using an iridium-based Crabtree-like catalyst for the anomerization of the allyl group followed by a iodine-promoted deprotection. However, these conditions have been previously successfully used by our group and others in many occasions (see Tamigney Kenfack, M. *et al. J. Org. Chem.* 2014, 79, 4615-4634; Gauthier, C. *et al. Org. Biomol. Chem.* 2014, 12, 4218-4232; Laroussarie, A. *et al. J. Org. Chem.* 2015, 80, 10386-10396). As suggested, more details regarding the transformation of disaccharide **34** into TCA derivative **35** have been added in the revised manuscript (see Results: Second generation synthesis of protected trisaccharides).

(2) In the binding studies, both disaccharides 6 and 7 show effective binding with mAb 4C7 in the ELISA and SPR assay (although the binding of 6 is indeed stronger), however, there is completely no binding of 7 in the STD-NMR experiments. The authors might comment on this. Additionally, the STD-NMR experiments show the strong involvement of the terminal talose residue in the binding with mAb, the authors might comment on if the appending 2-O-acetyl group also involves.

Response: The absence of signals in the STD NMR spectrum of the disaccharide **7** could be due to unfavorable binding kinetics considering that the ligand off-rate is extremely important for the overall sensitivity of the experiment. Given that the temperature strongly influences the kinetics and consequently the observed STD effects, we decided to run a STD NMR spectrum decreasing the temperature to 283 K (see revised Figure 7). Interestingly, under these experimental conditions, some slight, although not quantifiable, STD enhancements were observed for the mAb 4C7 – disaccharide **7** complex, confirming that the temperature might have affected the K_{off} .

In addition, as regards the contribution of the appending 2-O-acetyl group, we have further optimized the experimental conditions of STD NMR spectra in order to investigate the interaction between the disaccharide **6** and the monoclonal antibody. In the previous spectrum, the OAc-group of disaccharide **6** at the talose moiety was completely subtracted likely due to experimental conditions such as spin lock pulse applied to reduce the intensity of broad antibody resonances and/or antibody concentration. Thus, we acquired new STD spectra by using a lower concentration of the antibody (12 μ M vs 33 μ M) in deuterated phosphate buffer, a weaker spin lock (50 db vs 10 db) and a longer relaxation delay (4 sec). The resulting spectrum is reported in Figure 7 of the revised manuscript and shows a strong contribution of the acetyl group to the interaction, as expected.

(3) Page 7, chloroacetyl group should not be abbreviated as AcCl (that is acetyl chloride) but ClAc.

Response: This error has been corrected in the revised manuscript.

(4) Page S13, Supplementary Fig. 12, the linker should be five carbon long instead of four carbon.

Response: This error has been corrected in the revised supporting information file.

Reviewer #3 (Remarks to the Author):

This is an interesting study and expands on previous studies that demonstrated the protective efficacy of selected monoclonals. In this manuscript the authors go on to describe the recognised epitopes. From a biological prospective it is interesting and would guide further research and worthy of publication and likely to impact future vaccine design.

Response: We thank the reviewer for his/her positive comments regarding our work.

The strategy to link the antigens to the CRM is standard.

The authors demonstrated that the antibody responses recognised the OAg antigens. I would have like to have seen antibody binding to the whole bacteria - more biological relevance not just the antigens.

Response: Additional experiments and text have been added to the revised manuscript to address this issue (see Results: Immunization of mice with disaccharide- and OAg-based glycoconjugates - second paragraph, Supplementary Figure 175 and Supplementary methods). Using ELISA and immunofluorescence staining/microscopy techniques, we were able to confirm that **SOC-6** immune serum reacts strongly with both purified *Bm*-like OAg and paraformaldehyde-fixed *Bm*, respectively. Due to the inability of BALB/c and C57BL/6 mice to produce antibody responses against the terminal epitope of *Bp* OAg or disaccharide **7** (see Results: Immunization of mice with disaccharide- and OAg-based glycoconjugates - second paragraph and revised Figure 8 and Supplementary Figure 176), similar studies were not conducted with **SOC-7** immune serum.

Since *B. pseudomallei* is an intracellular pathogen I would have like to have seen some additional experiments and discussion on the cellular immune responses. Especially since the authors use PolyI:C adjuvant (with Alhydrogel) that would up-regulate innate immune responses through activation of TLR3 and subsequently up regulation of IL-12 and IFNs.

Response: Other than raising T-cell responses against the CRM197 carrier protein (which are critical for enabling high titer IgG responses to be produced against the covalently-linked haptens - disaccharides **6** and **7**), immunization of mice with **SOC-6** or **SOC-7** would not be predicted to elicit any protective cellular responses. As the Reviewer is aware, the main objective of immunizing with glycoconjugates is to stimulate protective humoral responses. This being the case, we do eventually plan to assess the functional activity of our immune serum samples (via opsonophagocytosis/opsonophagocytic killing assays) but only once we are able to optimize the immunogenicity of **SOC-6** (e.g. obtain more reproducible responses against disaccharide **6**) and find a suitable animal model that enables us to produce antibody responses against the terminal epitope of *Bp* OAg or disaccharide **7** (see Results: Immunization of mice with disaccharide- and OAg-based glycoconjugates - second paragraph and revised Figure 8 and Supplementary Figure 176). As for the PolyI:C/Alhydrogel adjuvant system, it was not our intention to use it to promote protective cellular immune responses. Instead, we used this adjuvant system to formulate our glycoconjugates since, in our experience, it enables us to generate higher titer IgG responses against oligosaccharides/polysaccharides than using Alhydrogel alone (presumably due to the production of IL-12 and IFNs as noted by the Reviewer).

On the NMR and SPR data I make the following statements:

6 and 7 seem both to have high affinity according to SPR but only 6 binds in the NMR experiment. This doesn't match with the fact that the second OAc group of 7 might abolish binding. It's still nanomolar.

Response: The absence of signals in the STD NMR spectrum of the disaccharide **7** in the presence of the mAb 4C7 could be due to unfavourable binding kinetics since the ligand off-rate is extremely important for the overall sensitivity of the experiment. To confirm the above, a slight increase of some STD signals intensity was observed for disaccharide **7** in the presence of the mAb 4C7 when the temperature was decreased to 283 K (see revised Figure 7).

I find it also intriguing that the only OAc-group of 6 at the talose moiety is completely subtracted and not interacting at all although the talose is strongest engaging residue. This could only happen if the OAc is completely solvent exposed.

Response: The *O*-acetyl group of disaccharide **6** at the talose moiety was completely subtracted likely due to experimental conditions like spin lock pulse applied to reduce the intensity of broad antibody resonances and/or antibody concentration. Thus, we acquired new STD spectra by using a lower concentration of the antibody (12 μ M vs 33 μ M) in deuterated phosphate buffer, a weaker

spin lock (50 db vs 10 db) and a longer relaxation delay (4 sec). The resulting spectrum is reported in the Figure 7 of the revised manuscript and it shows a strong contribution of the acetyl group to the interaction, as expected.

The authors should run the STD of **7** at 283 K and 310 K to see if there is a temperature effect. It's plausible though that a second OAc group might abolish binding.

Response: We have run the STD spectra of disaccharide **7** in the presence of the mAb 4C7 at different temperatures, 283 K, 298 K and 310 K (see revised Figure 7 and Supplementary Figures 171 and 172). As also added to the main text of the revised manuscript, no STD signals were observed at 298 K and 310 K. However, when the temperature was set at 283 K, some slight STD enhancements were observed for the mAb 4C7 – disaccharide **7** complex, indicating a temperature effect.

I don't understand how the authors determine 40% of the hydroxyl group at C3 of **6** ? (blue circle Figure 7). Did they not use D2O? This aspect doesn't make sense.

Response: We determined the percentage of the STD effect belonging to the *O*-methyl group at C3 of the talose residue. It is not a hydroxyl group.

What are the T1/T2/.... in Figure 7? Needs to be labelled in the structure.

Response: The proton resonances belonging to talose and glucose residues were indicated with letters, **T** and **G**, respectively. We have modified the caption of the Figure 7, indicating what the letters stand for.

All in all the paper presents a range of interesting data, but to be accepted in the journal additional studies would need to be performed.

Reviewer #4 (Remarks to the Author):

In their manuscript entitled "Deciphering minimal antigenic epitopes associated with Burkholderia pseudomallei and Burkholderia mallei lipopolysaccharide O-antigens", Kenfack et al. describe a synthetic approach to access glycan epitopes of the Bp and Bm LPS O-antigens. This is a highly interesting and relevant study that highlights the utility of synthetic carbohydrate chemistry to identify immunogenic glycan epitopes as candidates for novel glycoconjugate vaccines. The authors' approach combines synthetic carbohydrate chemistry with a detailed characterization of the recognition of the minimal glycan epitopes by monoclonal antibodies (mAbs). Finally, they perform immunization studies in mice to determine the immunogenicity of the minimal glycan epitopes. The data on the molecular interaction of the synthetic oligosaccharides with LPS-specific mAbs as determined by glycan array are convincing. The performed SPR and STD-NMR measurements to determine KD values and binding epitopes, respectively, allow for an in-depth insight into the crucial role of the talose moiety in the glycan/mAb interaction. The manuscript is well written and of interest to a broad readership.

Response: We thank the reviewer for his/her positive comments regarding our work.

While the strength of the manuscript is clearly the chemistry part and the biophysical characterization of the glycan/mAb interactions, its weakness is the murine immunization studies. In my opinion, the in vivo studies need to be extended and fine-tuned to render the manuscript acceptable for publication (see "Specific points"):

Specific points:

1.) Why was a steady-state affinity model applied to calculate the KD values from the SPR data (p.12) given that a mAb was used as analyte (i.e. a bivalent analyte)?

Response: In the Biacore data analysis package, there are two ways to assess an experiment like ours. The first is the kinetic method, which results in an estimate of both the association rate constant (k_a) and the dissociation rate constant (k_d). These values can then be used to calculate the dissociation constant ($K_D = k_d/k_a$). The second method is the concentration method, in which a plot of RU_{max} vs mAb concentration is constructed and the subsequent analysis results in another estimate of the K_D . This value is sometimes referred to as the steady-state K_D or the apparent K_D . Initially, both kinetics and concentration methods were used to evaluate the binding affinity between mAb 4C7 and the synthesized oligosaccharides in this study. The results derived from both methods showed that mAb 4C7 binds to immobilized **BIO-6** with a higher affinity as compared to **BIO-7**. We agree with the reviewer that assessment of the binding affinity using the thermodynamic constants ($K_D = k_d/k_a$) is the traditional way to do this and results in somewhat higher estimates of

affinity. However, in this study we chose to report steady-state K_D because it is more reliable as suggested by the control statistic parameters (standard deviation and χ^2 values). In addition, in much of the literature, the steady-state K_D is used for purposes of comparison (e.g., analysis of a mAb binding to modified targets). Also, we feel though that the inclusion of the k_a and k_d values may be of interest to readers who want to compare rates of association/dissociation or calculate the K_D by the other method on their own. Again, thank you for your careful consideration of our study.

2.) The authors state that "future work [should] include optimizing the loading of oligosaccharides onto carrier proteins, varying the dose of the glycoconjugates delivered and determining the most effective adjuvant system" (p.15, ll. 462). In my opinion, however, the present data do not (yet) convincingly show strong immunogenicity of the selected glycan candidates (and the glycoconjugates used for immunization respectively). A prime-boost immunization protocol was employed (three immunizations in total) and a combination of the adjuvants Alum and PolyI:C was used. Still, the obtained IgG titers were fairly low given a limit of detection (LOD) of 100 (as shown in Figure 8C). While IgG responses were generally low for immunization with SOC-7, SOC-6 elicited higher antibody titers that, however, varied among mice (low titers in two out of six mice). I suggest trying alternative immunization protocols and adjuvants to obtain more reliable data on immunogenicity. In addition, the number of immunized mice could be increased to obtain more precise results for the endpoint titers.

Response: Additional experiments and text have been added to the revised manuscript to address these issues (see Results: Immunization of mice with disaccharide- and OAg-based glycoconjugates - second paragraph and revised Figure 8). In Figure 8b, we now show that when immunized with OC-4744 (RR4744 OAg-CRM197), high titer antibody responses in 5/6 mice can be produced against RR4744 OAg as well as disaccharide **6** (terminal epitope of the OAg). This being the case, it is unlikely that the adjuvant system or the number of mice that we used for our studies can account for the variable immunogenicity of **SOC-6**. Instead, as previously suggested, we think that optimizing the loading/presentation of disaccharide **6** on CRM197 will help to resolve this issue. As for the poor immunogenicity of **SOC-7**, immunization of BALB/c and C57BL/6 mice with OC-2808 (RR2808 OAg-CRM197) suggests that we will be unable to generate antibody responses against the terminal epitope of RR2808 OAg or disaccharide **7** in these animal models (see revised Figure 8 and Supplementary Figure 176). As previously suggested, this appears to be due to a hole in their B cell repertoire since humans have the ability to generate antibody responses against the terminal disaccharide of *Bp* OAg (see revised Figure 8).

3.) Along the same lines: The authors determined cross-reactivity of the antibodies induced by immunization with SOC-6 and SOC-7 with purified *Bp* O-antigens (Figure 8C). While strong cross-reactivity was observed upon immunization with SOC-6 in at least three out of six immunized mice,

barely cross-reactive IgG responses were observed upon immunization with SOC-7. Still, even for SOC-6, there were generally marked reductions in IgG titers against the purified Bp O-antigens compared to the immobilized disaccharides (up to a half order of magnitude). What is the reason? For instance, have the authors determined antibody responses against the linker?

Response: The observation that 5/6 of the **SOC-6** serum samples did not exhibit equal levels of reactivity with disaccharide **6** and RR4744 OAg (which are two similar but non-identical target antigens) is not unexpected (see revised Figure 8a). While disaccharide **6** and the two terminal residues of RR4744 OAg are structurally identical to one another, in the context of an immune assay (e.g. ELISA), they are not displayed in the same manner (e.g. anchored by a small linker vs. a large polysaccharide chain) which may influence antibody binding to the target antigens resulting in differing levels of reactivity. This phenomenon is also observed with the mouse mAbs (Figure 5b), mouse immune serum (see revised Figure 8b and Supplementary Figure 176) and human immune serum (see revised Figure 8c) which further supports this explanation.

4.) Finally, it is well-known that LPS-specific monoclonal antibodies are protective in infection with Burkholderia and protection can be transferred by passive immunization (e.g. Trevino et al., Infect. Immun. 2006, 74, 1958; AuCoin et al., PLOS One 7, e35386, and other studies correctly cited by the authors). Since the potential of LPS-based subunit vaccines against Burkholderia has previously been shown, I consider a challenge study necessary in which the authors address the protective capacity of the glycoconjugates. Demonstrating the protective potential of CRM glycoconjugates would markedly strengthen the impact of the manuscript and would justify the authors' claim that "these studies serve as foundation for the development of novel therapeutics, diagnostics and vaccine candidates to combat diseases caused by Bp and Bm" as stated in the abstract (p.1).

Response: Immunization of BALB/c and C57BL/6 mice with OC-2808 (RR2808 OAg-CRM197) or **SOC-7** suggests that we will be unable to generate antibody responses against the terminal epitope of RR2808 OAg and disaccharide **7** in these animals (see Results: Immunization of mice with disaccharide- and OAg-based glycoconjugates - second paragraph and see revised Figure 8 and Supplementary Figure 176). Since these mouse strains are the most frequently used animal models of experimental melioidosis, we are unable to assess the protective capacity of **SOC-7** until we identify an alternative animal model. As for **SOC-6**, it would be unethical (from an animal welfare/IACUC perspective) to conduct a challenge study prior to optimizing the immunogenicity of the construct. This being the case, and with all due respect to the Reviewer, we do not believe that the inability to conduct these experiments at the present time lessens the overall quality or impact of our study.

REVIEWERS' COMMENTS:

Reviewer #3 (Remarks to the Author):

The authors have responded to the majority of my comments and I believe that the manuscript is in much better shape. My view is that the paper is suitable for publication in the journal, subject to further editorial consideration.

Reviewer #4 (Remarks to the Author):

In their manuscript entitled "Deciphering minimal antigenic epitopes associated with *Burkholderia pseudomallei* and *Burkholderia mallei* lipopolysaccharide O-antigens", Kenfack et al. describe a synthetic approach to access glycan epitopes of the Bp and Bm LPS O-antigens. This is a highly interesting and relevant study that highlights the utility of synthetic carbohydrate chemistry to identify immunogenic glycan epitopes as candidates for novel glycoconjugate vaccines.

The manuscript has been considerably improved and the revised version addresses my main concerns. Although an *in vivo* challenge study has not been performed, I can accept the authors' explanation that immunogenicity of the construct has to be optimized first.

Reviewer #3 (Remarks to the Author):

The authors have responded to the majority of my comments and I believe that the manuscript is in much better shape. My view is that the paper is suitable for publication in the journal, subject to further editorial consideration.

We thank the reviewer for his/her positive comments regarding our work.

Reviewer #4 (Remarks to the Author):

In their manuscript entitled "Deciphering minimal antigenic epitopes associated with *Burkholderia pseudomallei* and *Burkholderia mallei* lipopolysaccharide O-antigens", Kenfack et al. describe a synthetic approach to access glycan epitopes of the Bp and Bm LPS O-antigens. This is a highly interesting and relevant study that highlights the utility of synthetic carbohydrate chemistry to identify immunogenic glycan epitopes as candidates for novel glycoconjugate vaccines.

The manuscript has been considerably improved and the revised version addresses my main concerns. Although an in vivo challenge study has not been performed, I can accept the authors' explanation that immunogenicity of the construct has to be optimized first.

We thank the reviewer for his/her positive comments regarding our work.